# Nutritional Genomics in Nonalcoholic Fatty Liver Disease

**DOI:** 10.3390/biomedicines11020319

**Published:** 2023-01-23

**Authors:** Carolina Vasconcellos, Oureana Ferreira, Marta Filipa Lopes, André Filipe Ribeiro, João Vasques, Catarina Sousa Guerreiro

**Affiliations:** 1Laboratório de Nutrição, Faculdade de Medicina, Centro Académico de Medicina de Lisboa, Universidade de Lisboa, 1649-028 Lisboa, Portugal; 2Instituto de Saúde Ambiental, Faculdade de Medicina, Universidade de Lisboa, 1649-028 Lisboa, Portugal

**Keywords:** nonalcoholic fatty liver disease, nutrigenetics, diet, nutrition, PNPLA3, TM6SF2, PEMT, CHDH, polygenic risk score

## Abstract

Nonalcoholic fatty liver disease (NAFLD) is a common chronic condition associated with genetic and environmental factors in which fat abnormally accumulates in the liver. NAFLD is epidemiologically associated with obesity, type 2 diabetes, and dyslipidemia. Environmental factors, such as physical inactivity and an unbalanced diet, interact with genetic factors, such as epigenetic mechanisms and polymorphisms for the genesis and development of the condition. Different genetic polymorphisms seem to be involved in this context, including variants in PNPLA3, TM6SF2, PEMT, and CHDH genes, playing a role in the disease’s susceptibility, development, and severity. From carbohydrate intake and weight loss to omega-3 supplementation and caloric restriction, different dietary and nutritional factors appear to be involved in controlling the onset and progression of NAFLD conditions influencing metabolism, gene, and protein expression. The polygenic risk score represents a sum of trait-associated alleles carried by an individual and seems to be associated with NAFLD outcomes depending on the dietary context. Understanding the exact extent to which lifestyle interventions and genetic predispositions can play a role in the prevention and management of NAFLD can be crucial for the establishment of a personalized and integrative approach to patients.

## 1. Introduction

Nonalcoholic fatty liver disease (NAFLD) is the primary cause of liver disease worldwide and is characterized by the presence of more than 5% hepatic steatosis, or fat accumulation in the liver, not caused by heavy alcohol intake or infectious hepatitis [1,2]. NAFLD includes two subtypes: nonalcoholic fatty liver (NAFL), which is characterized by the presence of steatosis without inflammation, and nonalcoholic steatohepatitis (NASH), which is characterized by the presence of both steatosis and inflammation, with the potential to progress to cirrhosis and hepatocellular carcinoma [2,3,4,5].

Several environmental factors, such as obesity, type 2 diabetes, hyperlipidemia, and metabolic syndrome, have been linked to the development of NAFLD [4,6,7]. Lifestyle habits, such as physical inactivity and an unbalanced diet, particularly in cases of sedentarism and excessive caloric intake, are major risk factors for the development of NAFLD, as they contribute to insulin resistance, dyslipidemia, and adiposity, which are key factors in the pathogenesis of the disease [2,3,7,8]. High fructose diets may also potentially increase the risk of NAFLD, although the evidence in the context of a normocaloric diet is not yet conclusive [3].

In addition to environmental factors, there are several genetic and epigenetic factors that can contribute to the development of NAFLD [8]. The association between single nucleotide polymorphisms (SNPs) and liver conditions, namely NAFL, NASH, liver fibrosis, and hepatocellular carcinoma, has been increasingly studied in recent years. The dysregulation of lipid metabolism could be a critical factor in the development and progression of these conditions in a way that gene polymorphisms that are vital for this regulation could be involved [9].

Epigenetics involves changes in gene expression or silencing without changing the DNA sequencing [10]. These mechanisms can include DNA methylation and binding of microRNAs, and histone modifications, which have been shown to play a role in the development of NAFLD [10,11].

For instance, hypermethylation in the regulatory region of the patatin-like phospholipase domain-containing 3 (PNPLA3) gene was found to be substantially hypermethylated in patients with advanced NAFLD [12]. On the other hand, the specific binding of microRNAs can regulate gene expression and has been identified in many different diseases, including liver conditions [13]. Specifically, the most studied ones include miR-122, miR-34a, miR-33a/b, miR-155, miR-451, miR-375, miR-192, miR-27a/b, miR-24, miR-149, miR-21, and miR-185, which can alter pathways such as glucose metabolism, lipid metabolism, and inflammation [13].

Genetic and nutritional factors are closely related, as the foods we eat can impact our genetic expression, and our genetic identity can also influence our responses to diet, including ingestion, absorption, and metabolism [14]. The study of how diet affects gene expression is called nutrigenomics, an emerging area of nutrition sciences dedicated to studying how diet affects gene expression and the modifications in protein and metabolites resulting from it [15,16]. In contrast, the study of how specific dietary patterns and nutrition are impacted by individual genetic variations is called nutrigenetics, which tries to explain how specific dietary patterns and nutrition are affected by individual genetic variations [10]. Nutrigenetics considers individual differences in health status and risk of diseases, even when dietary patterns between two or more people are similar [15].

In the context of NAFLD, understanding the individual response to diet, which is highly dependent on DNA expression, can be useful for the prevention, treatment, and management of disease [8]. Overall, this review aims to provide a comprehensive overview of the current state of knowledge on the role of genetics and nutrition in NAFLD and to highlight the potential implications of this knowledge.

Specifically, we will elaborate on four specific polymorphisms: PNPLA3, TM6SF2, PEMT, and CHDH, and highlight the dietary patterns/nutritional factors that are associated with their SNPs. Although other genes, such as apolipoprotein C3 (APOC3), and gene choline kinase alpha (CHKA) also seem to play a role in the genesis and development of NAFLD, their connections with nutrition and dietary patterns are still missing [9,17].

By examining the current scientific evidence on the role of these gene polymorphisms in the development of NAFLD and the dietary patterns/nutritional factors that are associated with them, we hope to provide a clearer understanding of the complex interplay between genetics and nutrition in the development and management of this disease.

## 2. PEMT Gene

The phosphatidylethanolamine N-methyltransferase (PEMT) gene presents various polymorphisms, and its expression was inversely associated with the severity of NASH [18,19]. This gene catalyzes de novo synthesis of phosphatidylcholine (PC) through the methylation of phosphatidylethanolamine (PE), mainly in the liver [18,20,21]. The PEMT pathway for de novo PC synthesis is shown in Figure 1.

PC is the metabolite of the choline metabolism and a component of the phospholipids of very low-density lipoprotein (VLDL) cholesterol, being necessary for the hepatic secretion of TG from VLDL molecules avoiding lipid accumulation [6,18,22]. Therefore, reduced availability of PC and choline, whether due to insufficient intake in the diet or impaired endogenous production, can lead to fat accumulation in the liver. Hepatic steatosis is influenced by the balance between lipids from the diet or from conversion of glucose into lipids within the liver, and the secretion of lipids from the liver in forms of VLDLs [22].

This secretion depends on the synthesis of a lipid envelope composed of apoproteins and PC, through the choline/1-carbon metabolism [6,22]. Several genes and related pathways are involved in the choline/1-carbon metabolism and have been associated with increased risk of NAFLD and liver steatosis, including, methyl-tetrahydrofolate (MTHFR), choline dehydrogenase (CHDH), betaine-homocysteine methyltransferase (BHMT), PEMT, solute carrier family 44 member 1 (SLC44A1), PNPLA3, and ATP-binding cassette subfamily B member 4 (ABCB4) [22].

The MTHFR gene is involved in PC synthesis obtained either from choline-derived or 1-carbon pool-derived methyl-groups [22]. The CHDH, BHMT and PEMT also seem to be important factors for the pathway forming PC [22]. On the other hand, the gene product of SLC44A1 is required in choline transport into the hepatocyte and then into the mitochondria [22].

PC can have two different destinations in the body. Firstly, it can be used for the VLDL synthesis through its hydrolysis, involving the PNPLA3 [22]. Alternatively, it can be secreted in the bile using a protein encoded by the ABCB4 gene [22].

### 2.1. Interaction PEMT-Choline

Choline can be found in various dietetic sources, such as eggs and soy products, and is an essential nutrient for normal liver function maintenance [6]. A choline intake of less than 50 mg/day is considered deficient, and is shown to exacerbate intra-hepatic levels of fat by increasing the accumulation of lipids in the liver, as well as hepatic enzyme concentrations [6]. For that reason, a poor choline-content diet may contribute to the development of NAFLD, once it disturbs the previously mentioned choline/1-carbon metabolism [22].

Yu et al. observed an inverse association between higher choline intake and risk of fatty liver, independently of socioeconomic factors, lifestyle, and intake of other nutrients [6]. Nevertheless, this association was attenuated after adjustment for other metabolic diseases and BMI, remaining significant only in normal weight-women [6].

Fischer et al. aimed to assess choline requirements in healthy men and women and the clinical implications of the deficiency of this nutrient [23]. For this purpose, participants ate a standard diet containing 550 mg/70 kg/day of choline for 11 days and then submitted to another diet with less than 50 mg/70 kg/day of this nutrient for 42 days [23]. After the low-choline diet, 44% of premenopausal women developed reversible fatty liver and damage to hepatic and muscle tissues, in comparison to the majority of men (77%) and postmenopausal women (80%) [23].

Indeed, these results appear to be due to the estrogen-improved capacity to produce endogenous choline in premenopausal women since estrogens induce the PEMT gene in the liver [24].

These findings were consistent with those of Piras et al. [19]. Indeed, hepatic PEMT expression seemed to be significantly reduced in postmenopausal women with NASH when compared with those with normal liver histology [19]. This suggests that a decrease in PEMT expression in women with low estrogen levels may be a feature of liver dysfunction [19].

Postmenopausal women could then benefit from an individualized clinical and nutrition intervention. Nevertheless, more evidence is needed, especially addressing women in different NAFLD stages.

The PEMT gene presents two relevant polymorphisms that could benefit from dietary intervention: rs12325817 and rs7946.

#### 2.1.1. Polymorphism rs12325817

The polymorphism rs12325817 is found on chromosome 17 and consists of a G to C conversion [25,26].

Costa et al. aimed to characterize the susceptibility to develop liver and muscle damage, characterized as organ dysfunction, when fed with a low choline diet [27]. For that, humans received a diet containing an adequate intake of choline for 10 days (550 mg choline), and then fed a diet containing little choline for up to 6 weeks (<50 mg choline) [27].

Women carriers of the variant rs12325817, especially carriers of the C allele, presented significantly increased risk of developing organ dysfunction when dietary intake of choline was insufficient [27,28]. It is suggested that this effect is especially true for post-menopausal women with lower estrogen levels, being more sensitive to the SNP [27].

These results were also confirmed by Fisher et al., showing that choline requirements are increased in the presence of the variant rs12325817 [26]. Postmenopausal women had a higher dietary requirement for choline, when compared with premenopausal women [26].

Fisher et al. showed for the first time that postmenopausal women who received a treatment with estrogen have a decreased choline dietary requirements [26]. For postmenopausal women with one or more alleles for the SNP, only 40% of those treated with estrogen developed organ dysfunction following a low choline diet, whereas 100% of those treated with placebo developed these signs [26].

These results show the need for specific dietary advice and choline intake, considering sex, stage of the lifespan, and PEMT rs12325817 genotype.

#### 2.1.2. Polymorphism rs7946

The polymorphism rs7946, or V175M, consists in a mutation of loss of function, in which a G is replaced by an A in the exon 8 of the PEMT gene [18]. Song et al. found a relationship between this variant and a greater susceptibility to NAFLD when the genotype of DNA samples from patients with this condition were compared to controls without the disease [18].

Three possible genotypes for this gene were identified: GG (Val/Val), GA (Val/Met), and AA (Met/Met), in which the latter is the most frequent in patients with NAFLD [18]. Therefore, it was suggested choline deficiency, from diet or inhibited PEMT activity, could lead to the onset of NAFLD [18]. Surprisingly, Costa et al. found no association between low choline intake and NAFLD susceptibility [27].

Interestingly, Wu C. et al. recently observed that the plasma concentration of free choline was significantly higher in cases with hepatic steatosis than in controls, although the ingestion of this nutrient was similar in both groups [29]. Surprisingly, they also found an association between a high plasma choline concentration in combination with GG genotype of the PEMT rs7946 variant and increased risk of hepatic fat accumulation in patients with NAFLD [29].

### 2.2. Interaction PEMT–Vitamin E

Vitamin E is a lipid soluble vitamin presenting antioxidant characteristics that can be found in various plant-based sources, such as nuts and vegetable oils [30]. Due to these characteristics, vitamin E can be used in the clinical setting to improve NAFLD symptoms since the progression from steatosis to steatohepatitis is highly dependent on oxidative stress [31]. Vitamin E is secreted from the liver and distributed to the plasma and other tissues, mainly associated with VLDL [30].

Presa et al. conducted a study aiming to determine if the supplementation of 0.5 g/kg of vitamin E could, in PEMT −/− mice, minimize high-fat diet-induced steatosis of the liver and the progression to NASH [31]. The supplementation proved to have a beneficial impact on VLDL-TG secretion in comparison to mice without supplementation [31].

It was observed that hepatic TG content was lower in PEMT +/+ mice in comparison to PEMT −/− mice [31]. However, although vitamin E induced a slight reduction in hepatic TG levels in knockdown mice, this reduction was not statistically significant [31].

Nevertheless, the dietetic intervention with vitamin E decreased oxidative stress, inflammation, and fibrosis in the liver [31]. Vitamin E was able to reduce concentrations of meaningfully correlated biomarkers for the advancement of NASH in these mice, including ceramides, sphingomyelin, sphinganine, and sphingosine [31]. Specifically for 1-deoxy-ceramides, vitamin E was able to reduce the concentration by 50% [31]. All these biomarkers are types of sphingolipids, which interact with several pathways involved in insulin resistance, oxidative stress, inflammation, and apoptosis, all of which are linked to NAFLD [32,33].

These findings support the idea that administration of vitamin E seems to prevent the progression from steatosis to steatohepatitis in mice PEMT −/− [31].

## 3. PNPLA3 Gene

The PNPLA3 gene (also known as adiponutrin), is a member of the patatin-like phospholipase family of proteins, expressed primarily in liver cells called hepatocytes and hepatic stellate cells, as well as in the adipose tissue [34,35,36].

The gene exhibits a strong ability to hydrolyze triglycerides (TG) into free fatty acids (FFA) and glycerol [34]. The diacylglycerols (DAGs) produced from the hydrolysis of TG contain polyunsaturated fatty acids (PUFAs) that will be transferred by PNPLA3 to generate polyunsaturated PCs [34,37]. Therefore, PNPLA3 is a key regulator of lipid droplets in the liver cells.

Even though it is clear that PNPLA3 plays an important role in the regulation of lipid metabolism, its exact role in the development and progression of NAFLD is still being investigated. One variant has been shown to have specific relevance and play a major role in NAFLD development. Indeed, the PNPLA3 rs738409 variant is reported as a major determinant of hepatic fat content and was found to be significantly associated with NAFLD in the overall population [2,8,9,38].

This polymorphism represents a cytosine (C) to guanine (G) substitution that leads to an amino acid replacement from isoleucine (I) to methionine (M) at codon 148 (I148M), resulting in an overall loss of function of PNPLA3 protein [39]. Hepatology guidelines from the European Association for the Study of the Liver (EASL) refer to this polymorphism as “the best-characterized genetic association” between a genetic modifier and NAFLD [2]. The presence of the variant has been linked to an increased risk of hepatocellular carcinoma, and it is an independent risk factor of NAFLD with carriers having a greater risk of developing advanced liver fibrosis and cirrhosis, as well as severe steatosis [2,22,38,40].

The variant leads to the accumulation of PNPLA3 at the surface of hepatocellular lipid droplets and inhibits the activity of other lipases, reducing triglyceride (TG) turnover and dismissal [8]. The expression of this variant is modulated by factors such as diet, blood sugar, insulin levels, and obesity [8].

### 3.1. Interaction PNPLA3–Essential Fatty Acids

Essential PUFAs are a type of fat that are essential for human health and are involved in various physiological processes in the body. They can be found in certain foods, such as fatty fish and nuts. As discussed, PNPLA3 is involved in the breakdown and metabolism of triglycerides, playing a role in the regulation of PUFAs in the body.

It has been shown that PUFA n-6 and n-3 levels in the diet can impact the accumulation of hepatic polyunsaturated triglycerides, particularly in carriers of the PNPLA3 rs738409 variant [3,41]. It is interesting to note that unlike insulin resistance and obesity associated with NAFLD, in which saturated and monounsaturated triglycerides are predominant in the liver, hepatic triglycerides in NAFLD patients that carry this SNP present predominantly PUFA, when compared to noncarriers [1,3,4,42,43]. In fact, although the increase in lipid content in the liver is detrimental to liver health, research has shown that the composition of the lipid droplets may actually be protective against cardiovascular disease [1,3,4,42,43].

Furthermore, a study by Luukkonen et al. found that besides the higher levels of intrahepatocellular triglycerides, homozygous 148MM variant allele carriers also present a deficiency of polyunsaturated TGs in VLDL, in both the fasting state and postprandial period [44]. This higher retention of PUFA in the liver of the carriers may lead to a deficiency of PUFAs in the VLDL-TGs released by the liver [44].

Additionally, the authors observed that carriers of the PNPLA3 148MM variant had a lower ratio of polyunsaturated linoleate fatty acid (18:2) to saturated palmitate (16:0) in their plasma TG-VLDL after consuming equal amounts of these fatty acids in a meal, compared to noncarriers [44].

Santoro et al., found that the interaction between this variant and n-6/n-3 PUFAs affects hepatic content in obese young people [45]. PNPLA3 seems to have an indirect role in lipid hydrolysis, with the enzyme showing a higher affinity for unsaturated fatty acids and diets rich in these fats appearing to have benefits for NAFLD patients [40,45]. However, an excess of n-6 PUFAs in the diet can serve as a substrate for the production of new triglycerides (TGs) and slow the hydrolytic function of PNPLA3, leading to liver steatosis [45]. 

Another study by Villar-Gomez et al. investigated the association between various nutrient intake patterns and fibrosis severity in patients with histologically confirmed NAFLD [46]. It was found that a higher intake of n-3 PUFAs was associated with a lower risk of fibrosis, and this association remained significant after controlling for various confounders such as age, BMI, and type 2 diabetes mellitus [46]. This effect appeared to be particularly strong for carriers of the rs738409 G-allele [46].

There is also evidence that n-3 PUFA supplementation may decrease liver fat [40]. Kuttner et al. looked at the impact of short-term omega-3 fatty acid supplementation (1840 mg eicosapentaenoic and 1520 mg docosahexaenoic acid) on hepatic steatosis in carriers of the PNPLA3 rs738409 variant, but did not find significant changes in hepatic steatosis in patients with NAFLD who were homozygous for the PNPLA3 I148M risk allele [40].

In a 2020 study by Name et al., the effect of reducing n-6/n-3 PUFAs for 12 weeks on NAFLD in obese youth was investigated, and the results differed from those obtained in a previous study that only increased n-3 PUFAs through supplementation [40,47]. The intervention in the 2020 study was able to reduce the percentage of hepatic fat fraction only in those homozygous for the PNPLA3 rs738409 risk allele (GG), regardless of weight loss [47]. These findings support previous research and suggest that manipulating n-6/n-3 PUFA intake may be more important than just increasing n-3 PUFAs [45,47].

Therefore, the overload of n-6 PUFAs in the diet may lead to hepatic steatosis and inflammation in NASH, while higher intake of n-3 PUFAs may decrease the risk of fibrosis and liver fat in individuals with NAFLD, especially in carriers of the rs738409 G-allele [40,45,46]. The impact of short-term omega-3 fatty acid supplementation on hepatic steatosis in carriers of the PNPLA3 rs738409 variant was not found to be significant [40], but it seems that reducing n-6/n-3 PUFAs can reduce hepatic fat fraction percentage in individuals with the rs738409 risk allele, regardless of weight loss [45,47]. These findings suggest that manipulating the balance of n-6 and n-3 fatty acids ratio may be more important than increasing/decreasing n-3 or n-6 intake alone.

### 3.2. Interaction PNPLA3–Carbohydrates

As with essential fatty acids, the consumption of simple sugars has also been studied in relation to steatosis, and there are still many contradictory results [46,48,49,50,51,52]. There is mixed evidence on the effects of dietary carbohydrates on liver outcomes, but it was suggested that dietary patterns, including intake of sweetened beverages and vegetables, may interact with the PNPLA3 rs738409 variant to influence the severity of steatosis [46,48,51,52].

In 2014, Nobili et al. published a study in Italian children and adolescents that reported an interaction between dietary compounds and PNPLA3 rs738409 with the severity of steatosis [48]. Dietary patterns analyzed included the intake of sweetened beverages and vegetables [48]. A significant interaction between the PNPLA3 rs738409 variant and high-sweetened beverages consumption on steatosis severity was found, suggesting it increases its severity [48]. Interestingly, diets poor in vegetables seemed to attenuate the predisposing effect of this variant on steatosis severity [48]. Noticeably, it is possible that the potential negative effect found for diets high in vegetables may be due to the inclusion of starchy vegetables, such as potatoes, naturally containing higher amounts of total carbohydrates [48].

Vilar-Gomez et al. found that in patients with biopsy-proven NAFLD, those with significant fibrosis had significantly higher carbohydrate intake compared to those with no or mild fibrosis [46]. This association was particularly apparent in individual carriers of the G-allele of the PNPLA4 rs738409 variant [46].

On the other hand, a randomized controlled trial in Latino youth with obesity did not find any significant association between liver outcomes and reducing dietary intake of sugar [49]. These findings go in line with those of Morrill et al., who did not find any significant association between level of hepatic steatosis and overall intake of carbohydrate, total sugar, added sugar, or fructose [50].

Sevastianova et al. investigated the effects of weight loss on liver fat [51]. It was found that weight loss through a low-carbohydrate diet was effective in reducing liver fat content, with carriers of the rs738409 G allele (148M/M) experiencing a greater decrease in liver fat compared to carriers of the I allele (148I/I) [51]. The authors concluded that weight loss through a low-carbohydrate diet may be particularly beneficial for individuals with the 148M/M genotype [51].

One year later, Sevastianova et al. aimed to investigate the effects of a 3-week intervention of overfeeding with simple carbohydrates on liver fat and de novo lipogenesis (DNL) in overweight subjects [52]. The authors concluded that the intervention increased liver fat and stimulated DNL but the correlation between the increase in DNL and the increase in liver fat content during carbohydrate overfeeding only was observed in PNPLA3 148I/I carriers [52]. All the changes were reversible following a 6-month weight loss period [52]. It can be concluded that excessive simple sugar intake plays a role in the pathogenesis of NAFLD, but decreases in body weight result in considerable decreases in liver fat [52].

These findings suggest that a high carbohydrate intake may worsen the reduced capacity of individuals with this genotype to hydrolyze TGs in the liver, and that weight loss through a hypocaloric and low-carbohydrate diet may be helpful in reducing hepatic fat, particularly for PNPLA3 148 M/M carriers [51,52]. However, it should be noted that the effects found from a low carbohydrate intake cannot be separated from the effects of weight loss [51,52].

Based on the research described, the intake of carbohydrates and sugar may impact the severity of liver fat accumulation, particularly in individuals with the PNPLA3 rs738409 variant. However, the results of studies examining these effects are mixed. Some studies have found that high intake of sweetened beverages and high carbohydrate intake may worsen the reduced capacity of these individuals to hydrolyze triglycerides in the liver, leading to increased liver fat [46,48,51,52], while others have found no significant association between level of hepatic steatosis and overall carbohydrate, total sugar, added sugar, or fructose intake [49,50].

Further research is needed to fully understand the role of PNPLA3 and dietary carbohydrates in the development and progression of NAFLD.

### 3.3. Interaction PNPLA3–Isoflavones, Methionine, and Choline

In animal models of NAFLD, short-term dietary deficiency of methionine and choline induced severe forms of NASH [53,54]. However, there is limited information on the impact of this deficiency on the development and progression of NAFLD in humans [46]. Both methionine and choline might be required in hepatic mitochondrial β-oxidation and very low-density lipoprotein (VLDL) synthesis [53].

There have also been recent studies showing that the consumption of soy isoflavones, which have estrogen-like effects, may reduce the risk of NAFLD in animal models, by promoting hepatic fatty acid β-oxidation, regulation of hepatic NDL and insulin signaling [53,55,56]. Thus, it may have positive effects on lipid metabolism, insulin resistance, inflammation, and oxidative stress.

Recently, the intake of soy isoflavones has been associated with a delayed progression of NAFLD and attenuation of hepatic steatosis in humans [46]. A higher intake of these nutrients may lower the risk of significant fibrosis in individuals with histologically confirmed NAFLD, with the effects being once again more pronounced in carriers of the rs738409 G-allele [46].

Further research is needed, especially in humans, to fully understand the impact of these nutrients on NAFLD and their effect, either from diet or supplementation, on the progression and/or treatment of NAFLD.

## 4. TM6SF2 Gene

The transmembrane 6 superfamily member 2 (TM6SF2) protein plays a major role in both very-low-density lipoprotein (VLDL) hepatic release and intestinal lipid clearance [57]. The rs58542926 polymorphism in TM6SF2 involves the substitution of G with adenine (A) at nucleotide 499, leading to the replacement of glutamic acid with lysine (L) at amino acid residue 167 (E167K) in the TM6SF2 protein [58].

It is thought that this variant increases the risk of liver fat accumulation through the retention of lipids and impaired VLDL export [59,60]. A meta-analysis of 12 studies involving 2889 cases and 1561 control subjects found a significant association between the TM6SF2 rs58542926 polymorphism and NAFLD in the overall population [9], as it promotes liver injury and abnormal regulation of lipid metabolism [58]. Notably, this relationship was significant in pre- and post-menopausal women, but not in men [57]. The hepatology guidelines from the EASL also confirm that carriers of this variant have a higher liver fat content and increased risk of NASH [2].

### 4.1. Interaction TM6SF2–Fish Intake

A case-control study in a Greek population found that a diet rich in fish, fatty fish, and nuts was associated with a decreased risk of developing NAFLD, due to their antioxidant content such as MUFA, PUFA, vitamin E, polyphenols, and phytosterols [61]. In contrast, a diet high in refined starchy foods, fast food, sweet spreads and sugar, sauces, and fried food was found to increase the risk of developing NAFLD [61].

It was also observed that for individuals carrying the rs58542926 risk allele for the TM6SF2 protein, the risk of developing NAFLD was higher when fish intake was increased by one more serving per week [61]. This suggests that the TM6SF2 protein plays a key role in regulating postprandial lipidemia and that an increased intake of PUFAs may lead to an accumulation of triglycerides in carriers of the TM6SF2 variant [61]. However, more research is needed to confirm these findings and to fully understand this gene-diet interaction.

### 4.2. Interaction TM6SF2–Caloric Restriction

It is known that a calorie restriction intervention has positive effects on NAFLD and shows improvement in hepatic steatosis [62]. Krawczyk et al investigated whether carriers of the TM6SF2 variant would show the same benefits, despite their genetic profile [62]. It was found that the presence of this variant did not significantly influence these positive outcomes and improvement in hepatic steatosis of a 4-month calorie-restricted dietary intervention [62].

Hence, a personalized intervention should be considered in NAFLD patients, as this may counteract–at least to some extent–the effect originating from the presence of the TM6SF2 variant [62].

## 5. CHDH Gene

As previously mentioned, the CHDH gene is involved in the choline/1-carbon metabolism and seems to be important for the pathway forming PC. Indeed, CHDH is a flavin-dependent mitochondrial enzyme responsible for the irreversible oxidation of choline to form betaine aldehyde [63]. This reaction reduces choline availability to produce PC [63].

### 5.1. Interaction CHDH–Choline

It is likely that the CHDH gene expression affects dietary requirements for choline [27]. The gene is also associated with altered susceptibility to develop organ dysfunction on a low choline diet [27].

Since the CHDH gene is responsible for processing choline, variations in this gene may have an impact on aspects of choline metabolism. Two relevant polymorphisms were identified, in which carriers could benefit from dietary intervention: rs9001 and rs12676.

Research suggests that the CHDH rs9001 variant has a protective effect on susceptibility to choline deficiency, while the rs12676 variant is associated with an increased susceptibility to choline deficiency, indicating that these variants may have opposite effects on the activity of CHDH [63,64].

#### 5.1.1. Polymorphism rs9001

The polymorphism rs9001 of the gene CHDH consists in a replacement of A for C in codon 318 [27]. It has been shown a protective effect on carriers of the C allele on susceptibility to develop organ dysfunction when individuals were fed with a low choline diet [27]

#### 5.1.2. Polymorphism rs12676

The polymorphism rs12676 of the gene CHDH consists in a replacement of G for a T in codon 432 [27]. This SNP was not associated with the susceptibility to develop organ dysfunction associated with choline deficiency amongst all participants of the study [27]. However, for premenopausal women, it was suggested that the variant may increase dietary requirements of choline since 83% of heterozygous carriers of the variant developed organ dysfunction on a low choline diet, compared to 20% who did so without the risk allele [27]. This might be explained by the fact that the CHDH gene is under the control of an estrogen promoter [63].

## 6. Polygenic Risk Score

The polygenic risk score (PRS) is the sum of trait-associated alleles carried by an individual [65]. Therefore, it represents an individual’s genetic predisposition to develop a disease or an associated outcome [65].

As previously mentioned, the management and treatment of NAFLD involve dietary interventions [2,3]. A prospective study which included 1521 individuals with NAFLD aimed to evaluate the association between the adequacy of dietary patterns and changes in hepatic fat, as well as the relationship between individual shifts in diet quality and the PRS [66].

Two different diet quality scores were calculated, namely the Mediterranean-Style Diet Score and Alternative Healthy Eating Index (AHEI) [66]. The Mediterranean-Style Diet Score (MDS) measures adherence to a traditional Mediterranean diet, which emphasizes high consumption of fruits, vegetables, whole grains, fish, and healthy fats such as olive oil, while limiting red meat, processed foods and added sugars [67]. The Alternative Healthy Eating Index (AHEI) on the other hand is a measure of diet quality that assesses the overall quality of a person’s diet by scoring them based on their intake of 11 key food and nutrient groups such as fruits, vegetables, whole grains, nuts, and healthy fats, and their avoidance of red and processed meats, trans fats and sugary drinks [68].

Subsequently, the interaction between these results and the PRS considered for NAFLD was evaluated. The PRS was determined from a set of 5 SNPs from published Genome-Wide Association Studies (GWAS) of NAFLD: PNPLA3 rs738409, nCAN rs2228603, LYPLAL1 rs12137855, GCKR rs780094, and PPP1R3B rs4240624 [66].

A higher PRS was found to be associated with more significant liver fat accumulation in individuals whose MDS and AHEI adequacy scores decreased between baseline and follow-up [66]. This means that it may be possible to counteract the genetic predisposition to NAFLD if we at least keep the diet quality stable over time [66]. These associations were largely explained by a change in the intake of certain common foods and nutrients in both diets assessed, such as increased ingestion of fruits, vegetables, nuts, legumes, whole grains, eicosapentaenoic acid (EPA), and eicosapentaenoic acid (DHA), as well as decreased consumption of red meat and trans-fat [66].

We can therefore conclude that dietary stabilization or improvement may be critical for individuals at high genetic risk for NAFLD (Figure 2).

## 7. Discussion

As previously mentioned, NAFLD is the leading cause of liver disease, representing a serious health problem worldwide [8]. There is still no consensus regarding NAFLD treatment nor pharmaceutical intervention approved; the best clinical approach to the disease is still weight loss and improvement of insulin resistance [2,3,4,8].

Understanding the exact extent to which lifestyle interventions and genetic predispositions can play a role in the prevention and management of NAFLD can be crucial for the establishment of a personalized and integrative approach to patients. Dietary composition may influence metabolism, gene and protein expression and may alter risk factors that can be crucial to achieving the most effective clinical outcome possible. Therefore, it is vital to understand the effects of the avoidance and inclusion of certain foods and nutrients and their individual consequences in patients with genetic variations.

This article has reviewed the growing evidence of gene-nutrient interactions in the context of NAFLD. The main results here discussed addressing the relationship between genetics (gene and polymorphisms), nutritional aspects, and NAFLD are available on Table 1. Evidence already demonstrates a strong relationship between the increased hepatic fat content and disease development in the presence of certain PNPLA3 gene variants [38]. The extent to which dietary composition may lead to more successful clinical outcomes in carriers of PNPLA3 rs738409 risk polymorphism has also been demonstrated, especially the reduction of carbohydrate intake and weight loss [8,48,51,52].

Regarding the PEMT gene, several studies have demonstrated an increased risk of developing liver dysfunction and NAFLD when choline intake is deficient in carriers of the variants rs12325817 and rs7946 [18,26,27,28]. Considering that the enzyme PEMT is induced by estrogens, this risk is higher in postmenopausal women due to lower levels of the hormone [19,23,24,26,27,28].

However, considering the current scarcity of studies conducted in humans, more investigations are still needed, especially testing the relationship between NAFLD development and progression, CHDH, and its relationship with diet.

Deeply investigating other genes and their variants would help make the nutritional practice and disease guidelines in the context of fatty liver more individualized and tailored to every patient’s needs.

One example where more insights regarding the impact of diet and nutrition would be of great use is in the context of the gene apolipoprotein C3 (APOC3), located on chromosome 11q23, an area of strong connection with lipid metabolism [69]. APOC3 is a coding protein expressed in hepatocytes, which impairs lipolysis of TG-rich lipoproteins by inhibiting their hepatic uptake by remnant receptors and lipoprotein lipase [17].

Recently, numerous studies started to investigate the existence of an association between SNPs in the APOC3 gene and the susceptibility to the development of NAFLD, but its connection with nutrition and dietary patterns are still missing. Although the results were not always consistent, two polymorphisms were the most explored: rs2854116 and rs2854117 [9,17]. Between these, only the APOC3 rs2854116 polymorphism seems to be significantly associated with a higher risk of developing NAFLD [9,17].

The gene CHKA should also be further investigated. CHKA is an enzyme that catalyzes the conversion of choline to phosphocholine, which is the first step in the pathway responsible for de novo biosynthesis of PC [70]. Different SNPs of the CHKA gene were investigated in terms of susceptibility for NAFLD. Indeed, postmenopausal subjects who carried the T allele of CHKA rs6591331 were more likely to develop organ dysfunction when submitted to a poor choline diet [28]. In contrast, variants such as rs10791957, rs7928739 and rs2512612, showed a protective effect towards a deficient choline diet, for homozygous carriers of the effect allele [28]. However, since current studies assessed organ dysfunction, including liver and muscle damage, the extent to which these effects are seen in the liver alone could not yet be evaluated.

Another example where additional investigations could help and guide the nutrition practice is Lean-Nonalcoholic Fatty Liver Disease (lean-NAFLD). Lean-NAFLD is an increasingly recognized condition that develops in patients with a body mass index (BMI) below 25 kg/m^2^ [71]. These individuals, despite having a normal weight, may have underlying genetic predispositions that lead to the accumulation of fat in the liver, also known as fatty infiltration [71]. Underlying genetic predispositions may explain the fatty infiltration in the liver for lean-NAFLD patients and further investigations are needed to understand the specific causes and guide appropriate nutritional practices for this population.

Bale et al. investigated the association between the PEMT rs7964 variant and a higher risk of NAFLD and NASH development [72]. Interestingly enough, this study only found an association between the gene variant and a higher risk to develop lean-NAFLD, but not NAFLD [72].

These findings are interesting in the fields of genetics and NAFLD, as underlying genetic predispositions may explain the fatty infiltration in the liver for lean-NAFLD patients. However further investigations are needed to understand the interaction between dietary patterns and/or specific nutrients intake and the genetic predisposition for the condition. New findings may help guide appropriate nutritional practices for this population.

The current practice of dietitians and nutritionists is based on the average requirements of specific nutrients established for the general population. However, different individuals may require specific recommendations, especially when carrying certain SNPs. For that reason, it is necessary to hone the nutritional intervention considering available evidence.

Yuan et al. conducted a systematic review and meta-analysis that investigated the relationship between homocysteine, folate, and NAFLD [73]. The study also used Mendelian randomization (MR) to examine whether the observed associations between serum concentrations of homocysteine and folate with NAFLD are causal [73]. The study found a causal positive association between high serum levels of homocysteine and high risk of developing NAFLD [73]. It also found an inverse association between high serum folate levels and a reduced risk of NAFLD in the meta-analysis, but it was not statistically significant upon further examination using MR analysis [73].

The study suggests that interventions aimed at reducing homocysteine levels and increasing folate levels may be effective in preventing NAFLD [73]. Further research to understand the influence of diet and supplementation on these mechanisms, particularly in individuals who are genetically predisposed to high homocysteine levels or low folate levels, would be valuable. Additionally, it would be interesting to investigate potential genetic factors, such as specific gene expression or SNPs, that may play a role in these mechanisms.

In recent years, there has been a growing interest of consumers in nutrigenetic-based individualized and personalized nutritional care. In this scenario, a recent consensus report of the Academy of Nutrition and Dietetics assesses whether the inclusion of genetic testing data in personalized nutrition intervention could benefit nutrition-associated outcomes [74]. Interestingly, there was a significant reduction in body fat percentage when patients with NAFLD were subjected to gene-specific dietary care based on SNPs and associated with glucose and fat metabolism [74]. 

These findings support the further need for studies aiming to assess the influence on health and diet outcomes resulting from the incorporation of nutrition counseling based on genetic testing results conducted by nutritionists.

To improve practical evidence on the matter, the Academy of Nutrition and Dietetics encourages registered dietitians and nutritionists who use genetic testing data in their clinical practice to make records of these clinical cases in the Academy of Nutrition and Dietetics Health Informatics Infrastructure [74]. This could present itself as a solution to create and enrich research, turning clinical visits into high-quality research data.

In conclusion, nutritional genomics in the context of NAFLD is still a developing but growing field. Further investigation is still needed to understand to what extent lifestyle interventions can play a crucial role in the prevention and management of the condition. Research outcomes with the potential to be converted into specific clinical recommenda-tions could be beneficial for patients and increase their engagement with treatment.

## Figures and Tables

**Figure 1 biomedicines-11-00319-f001:**
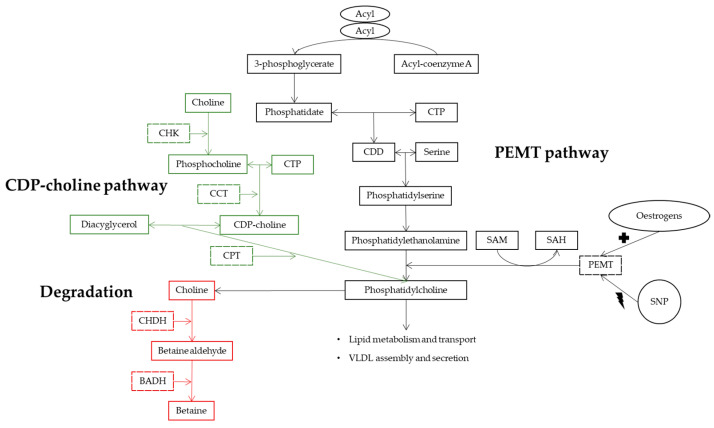
Choline metabolism and PEMT pathway for de novo phosphatidylcholine synthesis [21]. BADH, betaine aldehyde dehydrogenase; CCT, phosphocholine cytidyltransferase; CDD, cytidine diphosphate-diacylglycerol; CDP-choline, cytidine diphosphocholine; CHDH, choline dehydrogenase; CHK, choline kinase; CPT, choline phosphotransferase; CTP, cytidine triphosphate; PEMT, phosphatidylethanolamine N-methyltransferase; SAH, S-adenosylhomocysteine; SAM, S-adenosylmethionine; SNP, single nucleotide polymorphisms; VLDL, very low-density lipoproteins. Interpretation: There are two pathways for phosphatidylcholine synthesis, CDP-choline pathway (green) and PEMT pathway (black). The first one is the main pathway and begins with free choline being phosphorylated by CHK to phosphocholine. Then, phosphocholine reacts with CTP forming CDP-choline with this reaction being catalyzed by CCT. Finally, to originate phosphatidylcholine, the enzyme CPT esterifies CDP-choline with diacylglycerol. The PEMT pathway begins with acyl-coenzyme A donating 2 acyl groups to 3-phosphoglycerate which is converted to phosphatidate. Phosphatidate and CTP can react and form CDD, whose hydroxyl group can react with serine forming phosphatidylserine that can be decarboxylated to phosphatidylethanolamine. In liver, phosphatidylethanolamine can be methylated to form phosphatidylcholine which has an important role in VLDL assembly and secretion. The methylation of phosphatidylethanolamine is catalyzed by PEMT, which need three molecules of SAM to form one molecule of phosphatidylcholine, and it releases three molecules of SAH. PEMT can have its activity enhanced in the presence of estrogens or impaired by SNP that make PEMT at some level estrogens-irresponsive. If phosphatidylcholine synthesis is compromised the triacylglycerols from VLDL accumulate in liver and may lead to the onset of NAFLD. Phosphatidylcholine can also form choline and starts its oxidation to betaine by a process composed by two enzymatic reactions (red). The first one produces betaine aldehyde by CHDH, and the second one forms betaine by BADH.

**Figure 2 biomedicines-11-00319-f002:**
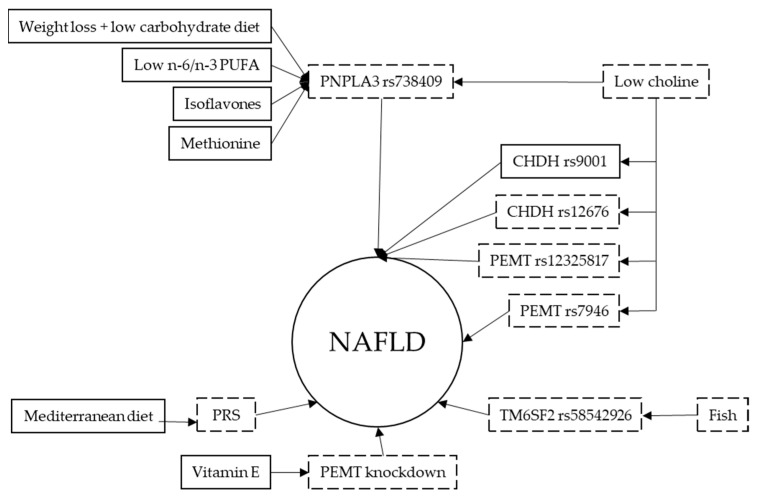
Nutritional genetics of NAFLD. Solid squares represent potential nutritional aspects and SNPs that positively affect NAFLD risk while non-solid squares represent nutritional aspects and SNPs that negatively affect NAFLD risk. Solid arrows indicate interaction while non-solid arrows indicate no interaction. PUFA-polyunsaturated fatty acids; PNPLA3-Patatin-like phospholipase domain-containing protein 3; CHDH-Choline Dehydrogenase; PEMT-Phosphatidylethanolamine NMethyltransferase; TM6SF2-Transmembrane 6 Superfamily Member 2; PRS-Polygenic Risk Score.

**Table 1 biomedicines-11-00319-t001:** Summary of studies reviewed that address the relationship between genetics (gene and polymorphism), nutritional aspects, and non-alcoholic fatty liver disease (NAFLD).

Gene	SNPs	Impact in Protein Function	Authors	Publication Year	Study Type	*N*	Nutritional Aspect	Main Results
PNPLA3	rs738409		Santoro et al. [45]	2012	Observational cross-sectional study	127 subjects	High dietary n-6/n-3 PUFA ratio	Positive interaction with hepatic fat content
			Kuttner et al. [40]	2019	Clinical trial	20 subjects	Short-term n-3 fatty acid supplementation	Did not significantly alter hepatic steatosis
		Reduced triglyceride hydrolysis capability	Vilar-Gomez et al. [46]	2021	Observational cross-sectional study	452 subjects	n-3 PUFA intake; Isoflavones intake; Choline intake; Methionine intake	Inversely associated with increased risk of significant fibrosis
							High-carbohydrate intake	Positively associated with increased risk of significant fibrosis
			Name et al. [47]	2020	Clinical trial	20 subjects	low n–6:n–3 PUFA ratio (4:1)	Amelioration of metabolic phenotype in adolescents with NAFLD
			Nobili et al. [48]	2014	Observational prospective study	200 subjects	High intake of sweetened beverages; High intake of vegetables	Significant positive interaction with steatosis severity in children and adolescents at risk of NAFLD
			Sevastinova et al. [51]	2011	Clinical trial	18 subjects	Weight loss through hypocaloric low-carb diet	Decreased liver fat
			Sevastianova et al. [52]	2012	Clinical trial	16 subjects	Short-term carbohydrate overfeeding and weight gain	Increased liver fat
			Schmidt et al. [49]	2022	Randomized controlled trial	105 subjects	Reduction of sugar intake	No improvements in liver outcomes, such as fat accumulation
			Morrill et al. [50]	2021	Observational cross-sectional study	288 subjects	Dietary intake	No significant association with levels of hepatic steatosis
TM6SF2	rs58542926	Distorted hepatic triglyceride secretion	Kalafati et al. [61]	2019	Case control study	351 subjects	High intake of fish, fatty fish, and nuts	Associated with decreased risk of developing NAFLD
							High intake of starchy foods, fast food, sweet spread, sugar, sauces, and fried food	Associated with increased risk of developing NAFLD
			Krawczyk et al. [62]	2016	Clinical trial	323 subjects	Caloric restriction	Variant did not influence the positive outcomes and improvement in hepatic steatosis of a calorie-restricted dietary intervention
PEMT	rs12325817	Reduced endogenous choline synthesis	Costa et al. [27]	2006	Clinical trial	57 subjects	Low choline intake	Significantly increased risk of developing organ disfunction, especially for postmenopausal women
			Costa et al. [28]	2014	Clinical trial	79 subjects	Low choline intake	Increased risk of developing organ dysfunction, especially for women carriers of the C allele
			Fisher et al. [26]	2010	Randomized controlled trial	49 subjects	Low choline intake	Increased choline requirements and risk of developing organ dysfunction, especially for postmenopausal women
	rs7946		Song et al. [18]	2005	Observational cross-sectional study	87 subjects	Choline deficiency	Associated with greater susceptibility to NAFLD
		Reduced endogenous choline synthesis	Costa et al. [27]	2006	Clinical trial	57 subjects	Low choline intake	Lack of effect in NAFLD susceptibility
			Wu et al. [29]	2022	Observational cross-sectional study	253 subjects	High plasma choline concentrations	Associated with increased risks of hepatic fat accumulation in patients with metabolic disease
CHDH	rs9001	N/A	Costa et al. [27]	2006	Clinical trial	57 subjects	Low choline intake	Protective effect on susceptibility to develop organ dysfunction
	rs12676	N/A	Costa et al. [27]	2006	Clinical trial	57 subjects	Low choline intake	Associated to susceptibility to develop organ dysfunction in premenopausal women

PNPLA3, patatin-like phospholipase domain-containing protein 3; n-6, Omega 6; n-3, Omega 3; PUFA, polyunsaturated fatty acids; TM6SF2, transmembrane 6 superfamily member 2; NAFLD, Non-Alcoholic Fatty Liver Disease; TG, triglyceride; MCD, Low methionine and choline diet; PEMT, phosphatidylethanolamine N-methyltransferase; CHDH, enzyme choline dehydrogenase.

## Data Availability

Not applicable.

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
