# Peer review of "Nutritional Genomics in Nonalcoholic Fatty Liver Disease"

_biomedicines, 2023, doi:10.3390/biomedicines11020319_

Round 1

Reviewer 1 Report

Carolina Vasconcellos et al discuss polymorphisms involved in NAFLD and nutritional factors and approaches that modulate genetic risks. The topic is very compelling and the concept could be truly applied in translational medicine. However, the manuscript is very confusing in its layout, many aspects are not explained well, and as it stands, it does not provide the reader a broad summary of major genetic susceptibilities in NAFLD that could be addressed with fairly positive results utilizing nutritional interventions.

·      In the introduction it is not clear what kind and which specific DNA methylation, histone modifications and microRNA patterns are observed in NAFLD. Moreover, when authors talk about genetic and nutritional factors, concepts are too vague and need to be expanded.

·      The structure of the review article is very confusing. It is not clear if authors intention is to elaborate on each described SNP and highlight dietary patterns associated. Moreover, authors should explain their rationale to describe either nutrigenetics or nutrigenomics associated with each SNP; they should possibly elaborate on both. This is an important point that authors need to address: they never truly discuss nutritional genetics and nutritional genomics aspects for each SNP /protein they consider.

·      Please describe in detail PNPLA3 function.

·      Paragraph about omega 3-6 is very confusing. When discussing PNPLA SNPs authors should make sure it is clear:

o   How are PUFA regulated physiologically by PNPLA3

o   How PUFA supplement or deprivation improve pathogenesis/outcome in NAFLD and in subjects carrying the SNP.

·      It is not clear if the SNP M/M is protective against accumulation of liver fat.

·      The data on association between low carbohydrate intake and steatosis appear very discordant between cited work. Please provide a consensus. Moreover, please specify when possible, amount of total carbohydrates vs sugars.

·      “weight loss” paragraphs could be integrated with the previous one.

·      Conclusions about paragraph 2.4 are not clear.

·      It is not clear what is the result in protein function from the SNP in TM6SF2 protein. Moreover patterns discussed in 3.1 need to be more detailed.

·      Paragraph 3.2 can be deleted as it does not provide any substantially new information from what is mentioned before. Same could be applied to paragraph 4.1.

·      How does increase in PC species inversely modulate NASH?

·      Please explain how poor choline diet enhances NAFLD in individuals with SNPs (and what kind of SNPs) in choline/1 carbon metabolism. Provide molecular mechanisms to sustain conclusions. 

·      How do authors explain the finding of higher free choline in cases of hepatic steatosis? (Ref 47)

·      Sentences regarding vitamin E need to be better contextualized. Moreover please specify which biomarkers dietetic intervention with vit E was able to ameliorate.

·      Introductions about the gene CHDH are very poor. Moreover authors talk a lot in paragraph 5 about choline and not much about PC. Since 5 and 6 interest similar pathways authors could merge the two and expand a bit the background on PC species in NAFLD.

·      Explain in more detail the MDS.

·      For table 1, it would be important to add a column regarding what the SNP does to protein function and a column concerning the molecular mechanisms explaining the main results.

·      It is not clear why APOC3 details are not mentioned in the table. Moreover, authors should better conclude the consequences of the APOC3 SNP speculating more about the available data. Is any APOC3 SNP associated with any disease at all?

·      Define CHKA.

·      Please introduce better the condition of lean NAFLD.

·      It would be important that authors expanded their figures: they could add small drawings for the main proteins they describe, highlighting with major functions and how dietary interventions affect related pathways in NAFLD.

·      Please provide more details about study in ref 56. It is not clear and it may appear contrary to the rationale the authors want to pursue the fact that in this study personalized dietary interventions based on genetics did not substantially improve NAFLD pathogenesis and progression. Please explain.

Author Response

Dear reviewers,

Thank you for taking the time to critically evaluate our submission and providing us with the opportunity to clarify and improve our work before publication. We have taken your feedback into consideration and have made the necessary revisions to the manuscript. We have addressed the areas of the manuscript that were lacking as per your comments and questions. We have included our detailed responses to the queries you had raised in the file attached.

We hope that the updated manuscript meets your expectations, and we would appreciate any further feedback you may have. Please do not hesitate to let us know if there are any other areas that can be improved.

We are grateful for the time and effort you have dedicated to reviewing our submission and look forward to receiving your feedback.

 Letter to reviewers

January 12, 2023

  1. REVIEWER 1

  1. « In the introduction it is not clear what kind and which specific DNA methylation, histone modifications and microRNA patterns are observed in NAFLD. Moreover, when authors talk about genetic and nutritional factors, concepts are too vague and need to be expanded. »

We better defined the concepts of epigenetics [lines 51-54] and nutrigenetics [lines 63-72]. For this purpose, we added new articles to support our explanations and we specified possible methylation and mRNAs that can be associated with NAFLD condition [lines 55-62].

  1. “The structure of the review article is very confusing. It is not clear if authors intention is to elaborate on each described SNP and highlight dietary patterns associated. Moreover, authors should explain their rationale to describe either nutrigenetics or nutrigenomics associated with each SNP; they should possibly elaborate on both. This is an important point that authors need to address: they never truly discuss nutritional genetics and nutritional genomics aspects for each SNP /protein they consider.”

The structure of the article was reworked in order to be presented in a clearer, less confusing fashion. Now, the structure goes as follows:

Abstract

Introduction

PEMT gene

PNPLA3 gene

TM6SF2 gene

CHDH gene

MBOAT7 gene

Polygenic risk score

Discussion

Specifically, we decided to start the article with the PEMT gene instead of the PNPLA3 gene, as it is involved in the choline/1-carbon metabolism, which is also important for other genes mentioned in the article.

Firstly, each gene has a brief description of its physiological role in the organism. Then, subtitles were added about the interaction of specific polymorphism and nutritional aspects. The texts under each subtitle contain a brief description of the impact of the polymorphism on the general activity of the gene.

In the introduction, we provided a clear outline and an explanation of the article’s purpose. The selection of polymorphisms was also justified: evaluate data on associations between the “Triad” Genes-NAFLD-Nutrition [lines 65-88].

  1. Please describe in detail PNPLA3 function.”

We included a description of the PNPLA3 gene and its physiological role in the organism, in regard of lipid metabolism. Then, like for the other genes, we specified the effect specific polymorphisms on PNPLA3 activity.

  1. Paragraph about omega 3-6 is very confusing. When discussing PNPLA SNPs authors should make sure it is clear:

D1) How are PUFA regulated physiologically by PNPLA3”:

We explained the interaction between PUFA regulation and PNPLA3 [lines 247-259].

D2)    How PUFA supplement or deprivation improve pathogenesis/outcome in NAFLD and in subjects carrying the SNP.”

We explained how dietary/supplemented PUFA are related to NAFLD in carriers of the SNP [lines 297-305].

  1. “It is not clear if the SNP M/M is protective against accumulation of liver fat.”

We are unsure how to understand this comment, but we confirm that SNP M/M consists in a mutation that positively benefits from a nutritional intervention.

  1. The data on association between low carbohydrate intake and steatosis appear very discordant between cited work. Please provide a consensus. Moreover, please specify when possible, amount of total carbohydrates vs sugars.”

It is not possible to provide a consensus, as the results of different studies show mixed evidence. We now made sure to specify that in our text and present a conclusion paragraph [lines 355-362].

Moreover, we modified the general sequence of approached concepts of this topic to facilitate comprehension. Specifically, we started to introduce observational studies [Reference 48, 50, 51, 52], followed by interventional studies [53, 54].

Unfortunately, it was not possible to specify the amount of total carbohydrates and sugars as this information is not given in the articles evaluated.

  1. “weight loss” paragraphs could be integrated with the previous one”

Weight loss paragraphs have been incorporated to the previous ones [lines 341 – 354].

  1. Conclusions about paragraph 2.4 are not clear.”:

Conclusions were restricted to facilitate comprehension and provide better explanation of the role of Isoflavones, methionine, and choline intake.

However, because we only cite one observational study, there is still no strong evidence to provide further explanations on the subject, as we now mention in our text.

  1. “It is not clear what is the result in protein function from the SNP in TM6SF2 protein. Moreover patterns discussed in 3.1 need to be more detailed.”

Result in protein function from the SNP in TM6SF2 protein in now specified [lines 389-391].

After careful consideration, we renamed subtopic 3.1 to “Interaction SNP-Fish intake”, as this was the only relatable pattern.

  1. Paragraph 3.2 can be deleted as it does not provide any substantially new information from what is mentioned before. Same could be applied to paragraph 4.1.”

Although the referred results in paragraph 3.2. are not very significant in general clinical setting, it is an interventional study that assesses the “triad” interaction: gene-NAFLD-nutritional aspect. For this reason, we believe that, if removed, we would in some way lose evidence that has been published. However, although we kept the study in our text, we simplified explanations to facilitate comprehension [line 415].

Regarding paragraph 4.1., it was reworked and rename to maintain relevance. The paragraph now focus on the interaction between MBOAT7 and methionine and choline, bringing new information to the study [line 481].

  1. “How does increase in PC species inversely modulate NASH?”

This information was added and further explained [lines 104-118].

  1. “Please explain how poor choline diet enhances NAFLD in individuals with SNPs (and what kind of SNPs) in choline/1 carbon metabolism. Provide molecular mechanisms to sustain conclusions. “

Information regarding choline deficiency was added [lines 121-127]. The general choline/1-carbon metabolism was further explained [lines 104-118].

  1. How do authors explain the finding of higher free choline in cases of hepatic steatosis? (Ref 47)”

No further data are available, because only the abstract was published. However, because it assessed the Gene-NAFLD-Nutrition triad, we have decided to keep this reference. 

  1. Sentences regarding vitamin E need to be better contextualized. Moreover please specify which biomarkers dietetic intervention with vit E was able to ameliorate.”

More context on vitamin E were added [lines 195-200]. Specific biomarkers mentioned by Presa et al. were also specified [lines 211-212].

  1. Introductions about the gene CHDH are very poor. Moreover authors talk a lot in paragraph 5 about choline and not much about PC. Since 5 and 6 interest similar pathways authors could merge the two and expand a bit the background on PC species in NAFLD.”

We decided to mention CHDH role in the paragraph about PEMT gene and choline [lines 105-113], because of their related pathways. Additionally, we provided an explanation of its general role in organism [lines 426-430].

  1. « Explain in more detail the MDS »

We further explained the MDS [lines 499-502]. The AHEI score was also exaplained, for better understanding and consistency [lines 502-506].

  1. « For table 1, it would be important to add a column regarding what the SNP does to protein function and a column concerning the molecular mechanisms explaining the main results. »

We added a column regarding what the SNP does to protein function.

However, we preferred to focus the content of the table on the Gene-NAFLD-Nutrition triad. We tried to provide further molecular mechanisms all along the text, whenever explanations are available by authors.

  1. « It is not clear why APOC3 details are not mentioned in the table. Moreover, authors should better conclude the consequences of the APOC3 SNP speculating more about the available data. Is any APOC3 SNP associated with any disease at all? »

APOC3 is now mentioned in the introduction. We specified the aim of this review to facilitate comprehension regarding genes and polymorphisms selected (only genes for which evidence exists on their interaction with diet/nutrition and NAFLD, would be included).

Nevertheless, some studies highlight the association of APOC3 SNPs and NAFLD, without clear relationship to diet. We explain these results in our discussion, to provide an opening for further investigation on the topic.

  1. « Define CHKA.»

We defined CHKA [lines 565-567].

  1. « Please introduce better the condition of lean NAFLD. »

We considered this comment and provided a better introduction on the condition of lean-NAFLD [lines 575-583].

  1. It would be important that authors expanded their figures: they could add small drawings for the main proteins they describe, highlighting with major functions and how dietary interventions affect related pathways in NAFLD. »

One major drawing with the main proteins, their dietary interventions, and related NAFLD pathways was included.

  1. « Please provide more details about study in ref 56. It is not clear and it may appear contrary to the rationale the authors want to pursue the fact that in this study personalized dietary interventions based on genetics did not substantially improve NAFLD pathogenesis and progression. Please explain.»

In fact, our paragraph on ref.56 was confusing. For that reason, we wrote the paragraph in a different way, to facilitate comprehension [lines 614-628].

We are grateful for the time and effort you have dedicated to reviewing our submission and look forward to receiving your feedback.

Sincerely,

Catarina Sousa Guerreiro

Reviewer 2 Report

Comments to Authors              

            This study showed that: a) the polygenic risk score represents a sum of trait-associated alleles carried by an individual and seems to be associated with NAFLD outcomes, depending on the dietary context; b) understanding the exact extent to which lifestyle interventions and genetic predispositions can play a role in the prevention and management of NAFLD can be crucial for the establishment of a personalized and integrative approach to patients.

          Authors are kindly requested to emphasize the current concepts about these issues in the context of recent knowledge and the available literature. This articles should be quoted in the References list.

References

  1. Physiopathology of nonalcoholic fatty liver disease: from diet to nutrigenomics. Curr Opin Clin Nutr Metab Care. 2022; 25 (5): 329-333. doi:10.1097/MCO.0000000000000859
  2. Plasma phospholipid arachidonic acid in relation to non-alcoholic fatty liver disease: Mendelian randomization study [published online ahead of print, 2022 Nov 6]. Nutrition. 2022;106:111910. doi:10.1016/j.nut.2022.111910.
  3. Homocysteine, folate, and nonalcoholic fatty liver disease: a systematic review with meta-analysis and Mendelian randomization investigation. Am J Clin Nutr. 2022;116(6):1595-1609. doi:10.1093/ajcn/nqac285.

Author Response

Dear reviewers,

Thank you for taking the time to critically evaluate our submission and providing us with the opportunity to clarify and improve our work before publication. We have taken your feedback into consideration and have made the necessary revisions to the manuscript. We have addressed the areas of the manuscript that were lacking as per your comments and questions. We have included our detailed responses to the queries you had raised in the file attached.

We hope that the updated manuscript meets your expectations, and we would appreciate any further feedback you may have. Please do not hesitate to let us know if there are any other areas that can be improved.

We are grateful for the time and effort you have dedicated to reviewing our submission and look forward to receiving your feedback.

  1. REVEIWER 2

  1. « Authors are kindly requested to emphasize the current concepts about these issues in the context of recent knowledge and the available literature. This articles should be quoted in the References list. »

All three recomended articles are now quoted in our review. Specifically:

  1. Physiopathology of nonalcoholic fatty liver disease: from diet to nutrigenomics. Curr Opin Clin Nutr Metab Care. 2022; 25 (5): 329-333. doi:10.1097/MCO.0000000000000859

[Lines 599-613]

  1. Plasma phospholipid arachidonic acid in relation to non-alcoholic fatty liver disease: Mendelian randomization study [published online ahead of print, 2022 Nov 6]. Nutrition. 2022;106:111910. doi:10.1016/j.nut.2022.111910.

[Lines 466-471]

  • Homocysteine, folate, and nonalcoholic fatty liver disease: a systematic review with meta-analysis and Mendelian randomization investigation. Am J Clin Nutr. 2022;116(6):1595-1609. doi:10.1093/ajcn/nqac285.

[Lines 36-41]

Sincerely,

Catarina Sousa Guerreiro

Round 2

Reviewer 1 Report

 Carolina Vasconcellos et al have provided an updated version of the manuscript that now appears well organized and of an easier read. They also have clarified the majority of points raised by the reviewer. The last aspect that needs to be taken care of regards the figures. At the comment of adding drawings for the main proteins, highlighting functions and how dietary interventions affect related pathways in NAFLD, authors replied to have provided a major drawing. However, drawing in fig 1 appears the same as before and is definitely not complete. It would be mandatory that authors answered to this point.

Author Response

Dear reviewer,

Thank you for taking the time to critically evaluate our submission and providing us with the opportunity to clarify and improve our work one more time before publication.

We have taken your feedback into consideration and have made the necessary revisions to figure 1, adding other metabolic pathways that are also part of choline metabolism, such as the CDP-choline pathway for phosphatidylcholine synthesis and one of the degradation pathways in which choline is oxidized into betaine. We think that figure 1 is now more complete and we hope we have understood your comment/suggestion correctly.

We are grateful for the time and effort you have dedicated to reviewing our submission and look forward to receiving your feedback.

Sincerely,

Prof. Dr. Catarina Sousa Guerreiro
